# Geographical origin determines responses to salinity of Mediterranean caddisflies

**Mauricio J. Carter**[1], **Matías Flores**[2], **Rodrigo Ramos-Jiliberto**[3]*

**1** Universidad Andrés Bello, Facultad de Ciencias de la Vida, Santiago, Chile, **2** Facultad de Ciencias, Universidad de Chile, Santiago, Chile, **3** GEMA Center for Genomics, Ecology & Environment, Faculty of Interdisciplinary Studies, Universidad Mayor, Santiago, Chile

* rodrigo.ramos@umayor.cl

## Abstract

Many freshwater ecosystems worldwide, and particularly Mediterranean ones, show increasing levels of salinity. These changes in water conditions could affect abundance and distribution of inhabiting species as well as the provision of ecosystem services. In this study we conduct laboratory experiments using the macroinvertebrate *Smicridea annulicornis* as a model organism. Our factorial experiments were designed to evaluate the effects of geographical origin of organisms and salinity levels on survival and behavioral responses of caddisflies. The experimental organisms were captured from rivers belonging to three hydrological basins along a 450 Km latitudinal gradient in the Mediterranean region of Chile. Animals were exposed to three conductivity levels, from 180 to 1400 $\mu S/cm$, close to the historical averages of the source rivers. We measured the behavioral responses to experimental stimuli and the survival time. Our results showed that geographical origin shaped the behavioral and survival responses to salinity. In particular, survival and activity decreased more strongly with increasing salinity in organisms coming from more dilute waters. This suggests local adaptation to be determinant for salinity responses in this benthic invertebrate species. In the current scenario of fast temporal and spatial changes in water levels and salt concentration, the conservation of geographic intra-specific variation of aquatic species is crucial for lowering the risk of salinity-driven biodiversity loss.

## Introduction

Increases in water salinity affect freshwater ecosystems in many regions of the globe [1, 2]. These changes in ionic concentration and composition impair water quality, which disrupts the abundance and distribution of aquatic species [1, 3–8]. This disruption threatens ecosystem functioning and the provision of services of crucial importance for our society such as drinking and irrigation water, climate regulation, recreation and food [9]. Mediterranean regions are specially vulnerable to this stress due to their marked seasonality in environmental temperature and evaporation.

To date, our understanding about the physiological mechanisms by which freshwater organisms deal with salinity stress is limited [2, 10]. Moreover, our knowledge of the

**Funding:** MCM was funded by grant CONICYT/
FONDECYT 1140498. RR-J was funded by grants
CONICYT/FONDECYT 1150348 and 1190173.

**Competing interests:** The authors have declared
that no competing interests exist.

consequences of salinity changes for organisms performance is relatively poor, considering the attention given to other stressors in freshwaters, such as metals and pesticides [11]. Salinity changes may exert direct lethal effects if those changes are strong enough as to exceed the tolerance limits of organisms [10]. Nevertheless, if salinity changes are more subtle, sublethal effects may be expressed such as alterations in development, physiology or behavior [12–17]. In particular if behavior is affected, this would imply a dysfunction in the short-term behavioral capabilities of reacting to environmental signals and stimuli. In benthic freshwater invertebrates, behavioral responses allow them to perform habitat selection, through deciding either to stay in the same place, to perform small—scale movements or to release from the substrate to derive and move away quickly [7, 18]. These behavioral responses to environmental risk are vital for organism performance and population persistence.

Freshwaters from Mediterranean regions are of special concern for biodiversity maintenance and ecosystem functioning. This is due to that Mediterranean ecosystems, on one hand, exhibit high levels of biodiversity and endemicity and, on the other hand, are subjected to high levels of biological invasions and are expected to be highly affected by climate change [19]. Rivers from Mediterranean Chile are characterized by strong environmental gradients [20–22], where both altitude and latitude are strong predictors of abiotic conditions.

Besides this, hydrological basins in this region have been threatened by economic activities mainly derived from mining and agriculture industry during the last century. These actions have disrupted physical, chemical and biological properties of these ecosystems [4, 23]. Thus, both natural –physical and geological– features as well as anthropogenic factors have contributed to create contrasting salinity levels in rivers belonging to the Mediterranean region of central Chile that extends from 31˚S to 35˚S along the Andean hydrological basins.

Physiological adaptation to different levels of water salinity has been reported for aquatic freshwater organisms, mainly linked with mechanisms of ion regulation [11, 24]. Thus, organisms from different local populations of a species may exhibit different degrees of salinity tolerance, which could also be expressed into diverse abilities to respond behaviorally to external stimuli. Nevertheless, due to our limited understanding of the ecological consequences of osmotic stress, it is unclear how the phenotypic responses to different salinity levels are shaped by local adaptation.

In this study we evaluate experimentally the effects of geographical origin and local salinity levels on survival and behavioral responses of a model benthic freshwater species inhabiting the Mediterranean region of Chile, the caddisfly *Smicridea annulicornis*. Our general prediction is that survival and behavioral responses to different salinity levels will be function of the geographical origin of the organisms. However, we have specific expectations according to two competing hypotheses. Habitats with lower water salinity represent a challenging environment for benthic invertebrate species, due to difficulties for osmotic regulation in a medium with a lower ionic concentration than internal fluids [25]. Thus, it has been suggested that the ability of some invertebrates to tolerate dilute water may allow them to inhabit a wide range of water salinity. However, physiological mechanisms to overcome low water salinity likely evolve at the expense of other traits that are beneficial at higher water salinity conditions [11, 26, 27]. According to this hypothesis, we expect that organisms adapted to low salinity will maintain similar levels of survival through different experimental salinity levels, but will reduce their behavioral performance at higher salinity as an expression of the costs of low-salinity adaptation. Also, we expect that organisms adapted to higher salinity levels will reduce their survival under low salinity. On the other hand, several studies indicate that freshwater organisms are negatively affected by increased salinity [28, 29]. According to this hypothesis, we would expect that low-salinity adapted organisms will be more affected by higher salinity levels both in behavioral responses and survival, as compared to organisms adapted to higher salinity

levels. Under this hypothesis, low-salinity adapted organisms should be more sensitive to experimental changes in salinity.

Our analyses shed light into how benthic freshwater fauna may respond to current salinity changes by evaluating an array of lethal (mortality) and nonlethal (behavioral responses) attributes.

## Materials and methods

### Study organisms

We conducted this study using as a model species the Neotropical trichopteran *Smicridea annulicornis* (Blanchard 1851), which has a broad geographic distribution in latitude and altitude [5]. The larval stages of this species inhabit the rhithral areas of rapids and pools where they are abundant and play a functional role as filter—feeders/collectors, consuming coarse organic particles and algae. During spring, large numbers of adults emerge and fly away [13, 30]. However genetic analyses indicated a low vagility of this species [31]. We collected the animals from three hydrological basins along a 450 km latitudinal gradient in the Mediterranean zone of central Chile: Choapa (S31˚ 52'; W70˚ 57'), Maipo (S33˚ 22'; W70˚ 27') and Maule (S35˚ 33'; W7'1˚ 13') ([Fig 1]). Biological samples were collected from free access riparian areas in Chilean rivers (DL N˚ 1.939-1977). In these areas, no permits are required for sampling invertebrate animals. In each of these localities, larvae of *S. annulicornis* were collected by hand from the riverbed in the spring-summer of 2016, and placed together in plastic jars of 1L to be transported to the laboratory. All animals were kept in water from the same place where the animals were collected and they were acclimated to the experimental conditions during one week before performing the measurements. We used electric conductivity as proxy of salinity. To determine the levels of conductivity to be used in the experiments we considered the range of variation of water conductivity at each of the hydrological basins reported for the last 50 years (Dirección General de Aguas, Chile; www.snia.dga.cl). We choose three experimental conductivity levels: low (180 $\mu S/cm$), medium (500 $\mu S/cm$) and high (1400 $\mu S/cm$). These levels were close to the ones measured by us *in situ* and approximate the conductivity values found historically at Choapa, Maipo and Maule basins respectively. Then, organisms from each population were randomly transferred individually into 40 mL glass jars to each experimental conductivity level for conducting the factorial experiment. Different salinity levels of the experimental media were prepared with aquarium sea salt (Sera$^{®}$) dissolved in distilled water up to reach the appropriate conductivity values, measured with a desktop electrical conductivity meter (Bante Instruments$^{®}$). The media were renewed every two days during the experimental period to avoid oxygen depletion and changes in pH. These parameters were recorded in parallel during the experiment.

### Measurement of response variables

During the experiment, all animals were kept in dechlorinated water, maintained at $20˚C$ with a photoperiod of 16L:8D. Animals were acclimated during 48 hours prior experimentation under the same ambient conditions used during measurements. To avoid further variability due to food conditions and to strengthen the responses to experimental factors, animals were kept unfed during the experiment. To conduct our experiment, we used 85 larvae from Choapa basin, 81 from Maipo basin and 48 larvae from Maule basin, distributed uniformly over the three conductivity levels (low, medium, and high. The body size (length from the top of the head to the end of the tail) of all experimental organisms was measured from digital images obtained with a digital camera attached to a stereo—microscope. Individual survival in each treatment was measured as the time from the beginning of the experiment to the death of

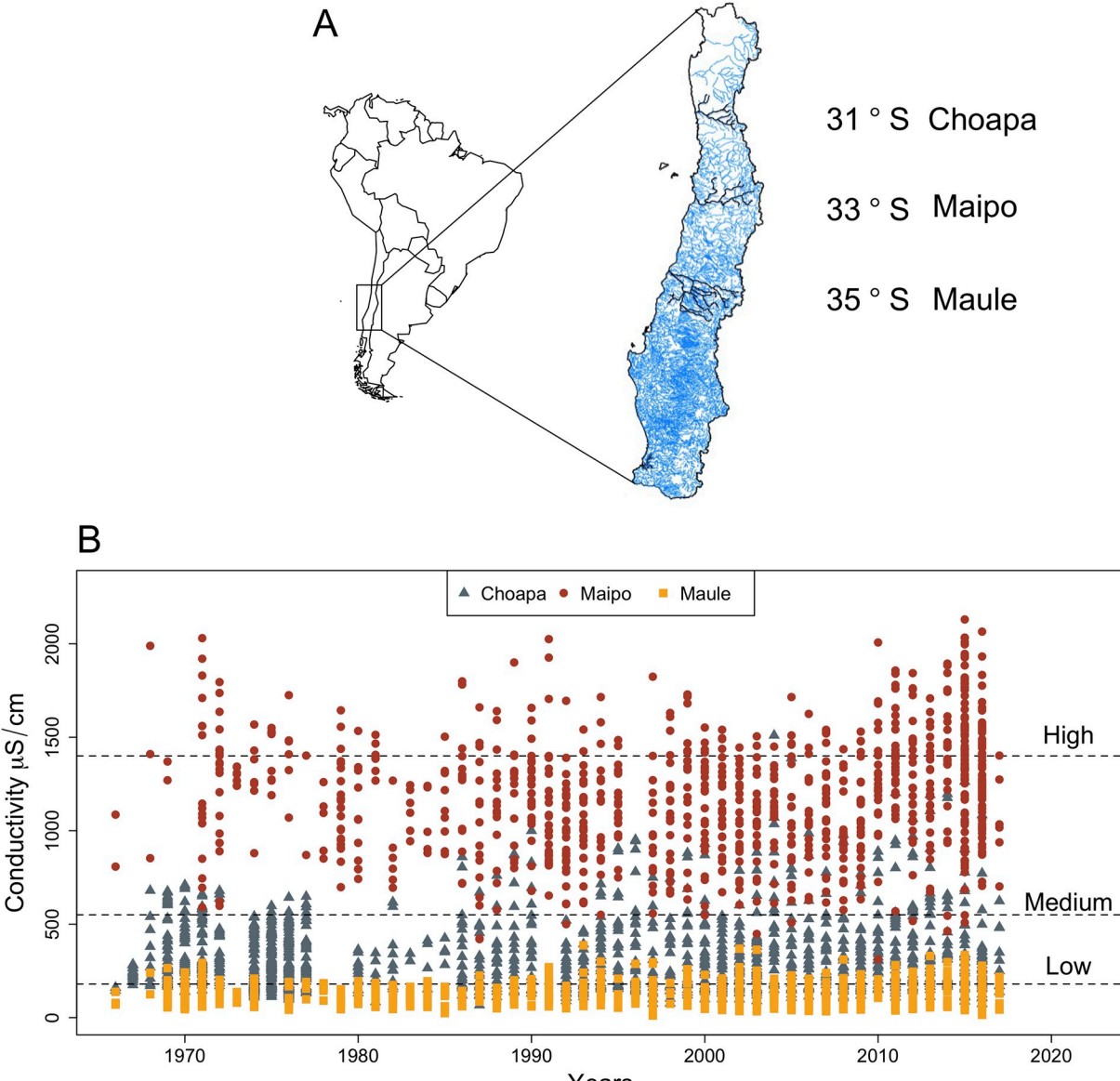

**Fig 1. Main features of the study sites.** A: Geographical location of the hydrological basins Choapa, Maipo and Maule in central Chile. B: Values of electric conductivities recorded for the three hydrological basins by the DGA (Dirección General de Agua, Chile; http://snia.dga.cl/ BNAConsultas/reportes) at the monitoring water quality stations during the last 50 years. Horizontal dashed lines indicate the three experimental conductivity levels used in our study.

the animal, after which the organisms were kept individually in 70% ethanol to measure body size. The behavioral traits were defined as the response to direct stimuli, reflecting the capability of responding to an external disturbance. To trigger the behavioral responses we used a pencil brush, with which we gently touched each animal when it was in a resting state. We defined four response traits from this procedure: i) "swimming" shown as an active movement through the water column; ii) "sheltering", characterized by assuming a spherical protection posture; iii) "walking" characterized by a regular displacement on the bottom of the glass using the front legs; and iv) "pushing-up" is characterized by a strong dorsal contraction of the body, forming an arched posture, at the same time that the animal sinks in the water column. We

recorded the occurrence of each behavioral trait within a time window of 30 seconds of observations per day during the entire experiment (≈ two weeks). Because we allowed the animals to perform any of these behaviors in each time window of observation, we considered the expression of each behavior responses as independent.

## Statistics

A factorial design was used to analyze the recorded responses. Our design included three levels of river basin and three levels of water conductivity. Survival analysis was conducted using a parametric model based on the Weibell distribution, with body size used as a covariate. For each trait, repeatability was evaluated by intra-class correlation coefficient [32] by calculating the between– and within—individual variance component, from models which consider the biased distribution of counts and proportional data [33]. We conducted the repeatability analysis using the rptR library in R software. The exhibition of any given behavior, here referred as "activity" was analyzed using a model which included body size, the time at measurement as well as the kind of behavioral trait adopted in each activity event recorded at each experimental condition. Because the behaviors were recorded as the presence or absence of the responses in discrete time (count data), our variable responses swimming, sheltering, pushing-up and walking were biased. Thus, to analyze the behavioral response variables, we fit Generalized Linear Models (GLMs) with error family according to this characteristic (quasibinomial). Body size and measurement times were used as covariates into the models to avoid under—sub estimating the behavioral responses to the combination of stress conditions. All the analyses were performed using R software [34].

## Results

### Survival time

Survival of the caddisflies *Smicridea annulicornis* differed significantly among organisms from different river basin. In addition there were nearly significant effects of conductivity (Table 1). The interaction between basin and conductivity did not exert significant effects on survival. The median survival time (time to 50% survival) for low, medium and high conductivity levels respectively, was 349 ± 45, 385 ± 24 and 197 ± 47 hours in caddisflies from Maule, 235 ± 26, 317 ± 22 and 214 ± 22 hours in caddisflies from Choapa, and 347 ± 29, 329 ± 33 and 349 ± 33 hours in caddisflies from Maipo (Fig 2). A visual analysis of the survival curves (Fig 2) suggests that conductivity level exerted: a) stronger effects on survival in organisms coming from Maule basin, with marked lower survival at high conductivity, b) lighter effects in caddisflies from Choapa, with higher survival at medium conductivity, and c) negligible effects in caddisflies from Maipo. These differences in organisms' sensitivity to conductivity is more evident when considering the variance of median survival time among the three conductivity levels

**Table 1. ANOVA results from survival parametric analyses with Weibull distribution to three caddisfly populations (Choapa, Maipo and Maule) exposed to low (180 μS/cm), medium (500 μS/cm) and high (1400 μS/cm) conductivity levels, body size was used as a covariate.**

| Factor | −2* Log Likelihood | χ—value$_{(df)}$ | P-value |
|---|---|---|---|
| Null model | 2760.77 | | |
| Body size (mm) | 2758.85 | 1.92$_{(1,207)}$ | 0.165 |
| Basin | 2751.66 | 7.19$_{(2,205)}$ | 0.027 |
| Conductivity (μS/cm) | 2746.67 | 4.98$_{(2,203)}$ | 0.082 |
| Basin x Conductivity | 2741.12 | 5.55$_{(4,199)}$ | 0.235 |

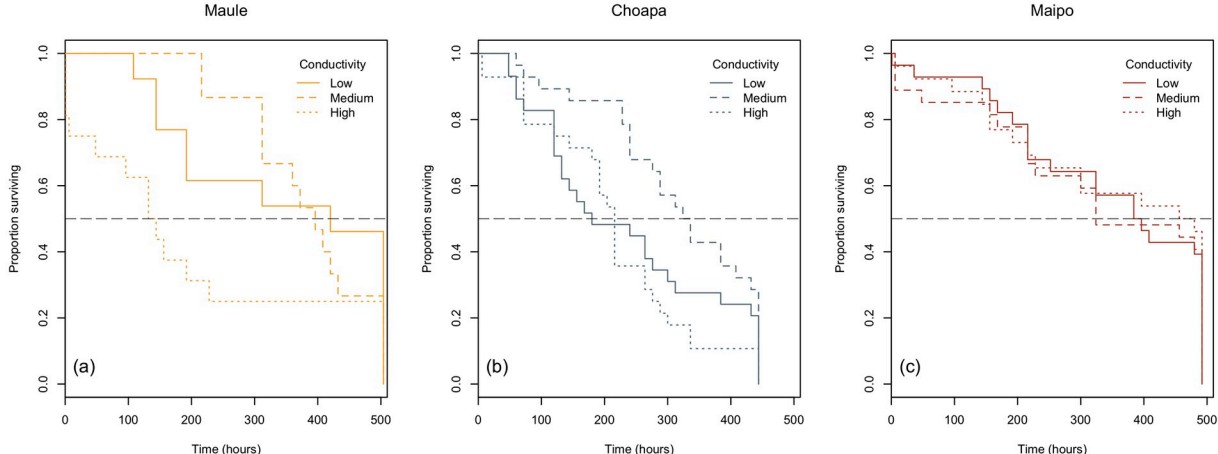

**Fig 2. Survival curves.** Kaplan-Meir survival curves of *S. annulicornis* from populations belonging to Choapa, Maipo and Maule river basins subjected to conductivity treatments. Doted black line indicate the 50% of survival.

within each basin: 9957.3 (Maule), 2962 (Choapa), and 121.3 (Maipo). The patterns describe above indicate intrinsic phenotypic differences among the tested populations.

## Behavioral responses

Activity (i.e. the rate of behavioral responses) was significantly affected by experimental time and body size (Table 2). In addition, organisms from different river basins showed significant differences in their activity levels in dependence on the specific behavior displayed (Table 2, Fig 3). In average, activity was lower in organisms from Choapa than Maule and it was the highest in organisms from Maipo. However, organisms from Maule basin exhibited lower levels of swimming behavior than organisms from both Maipo basin (Tukey test, $P < 0.001$) and Choapa basin (Tukey test, $P < 0.05$). Conversely, organisms from Maule basin exhibited higher levels of pushing-up than organisms from both Maipo (Tukey test, $P < 0.001$), and Choapa (Tukey test, $P < 0.001$). Pairwise comparison between Choapa and Maipo basins revealed lower levels of pushing-up in organisms from Maipo (Tukey test, z-ratio = 2.84, $P < 0.05$). Specific comparisons within Maule basin revealed that total activity as well as

**Table 2. Summary of results of ANOVA analyses with general linear model (GLMs) for total activity recorded from three caddisfly population (Choapa, Maipo and Maule) exposed to low (180 μS/cm), medium (500 μS/cm) and high (1400 μS/cm) conductivity levels.** Body mass and experimental time were used as covariates. Behavior categories were used as a factor to test if the different behaviors (swimming, sheltering, walking and pushing-up) differed in proportion among experimental treatments.

| Factor | F-value$_{(df)}$ | P-value |
|---|---|---|
| Experimental time (days) | $990_{(1,826)}$ | <0.001 |
| Body Size (mm) | $22.2_{(1,825)}$ | <0.001 |
| Behavior Category (BC) | $159_{(3,822)}$ | <0.001 |
| Basin | $3.78_{(2,820)}$ | 0.021 |
| Conductivity (μS/cm) | $0.07_{(2,818)}$ | 0.923 |
| BC x Basin | $15.5_{(6,812)}$ | <0.001 |
| BC x Conductivity | $0.87_{(2,806)}$ | 0.513 |
| Basin x Conductivity | $1.62_{(1,802)}$ | 0.166 |
| BC x Basin x Conductivity | $1.49_{(2,790)}$ | 0.123 |

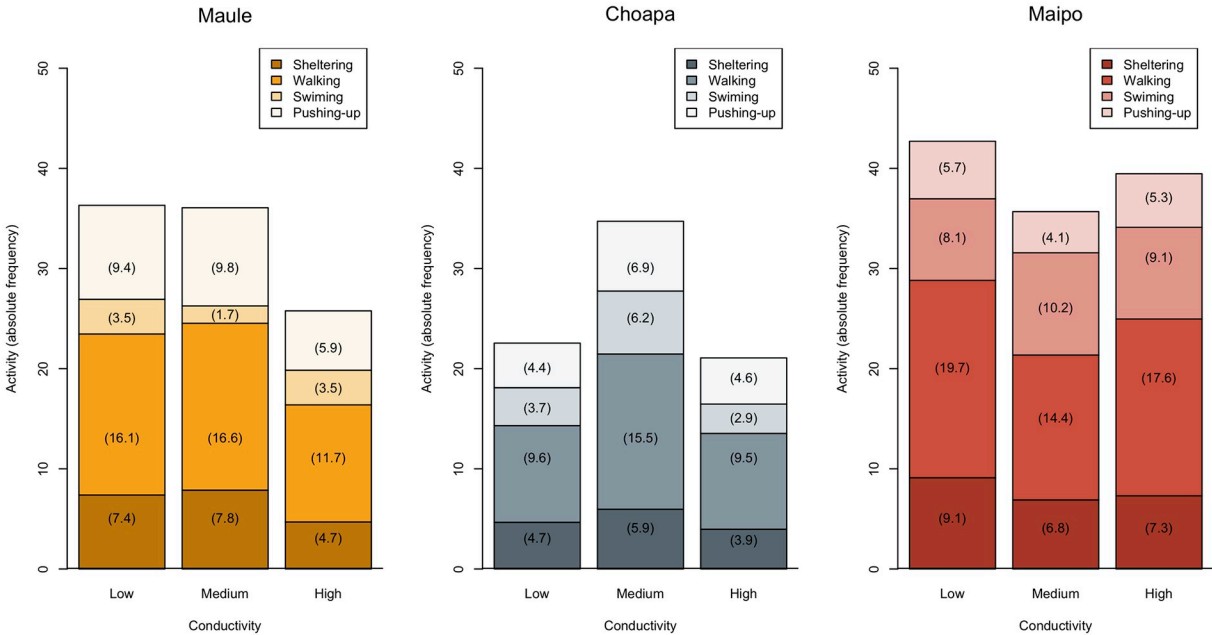

**Fig 3. Behavioral responses.** Mean total activity and its decomposition into specific behavioral responses exhibited by *S. annulicornis*. Values of activity levels are shown in parenthesis. Organisms came from three river basins: Choapa, Maipo and Maule. Experimental conductivity levels were: low (180 $\mu$S/cm), medium (500 $\mu$S/cm) and high (1400 $\mu$S/cm).

sheltering, walking and pushing-up behaviors diminished at high as compared to both low and medium conductivity levels, although these differences were not significant (Tukey test, $P > 0.05$). In Choapa basin, all behavioral traits exhibited higher activity rates at the medium conductivity level. These differences were statistically significant ($P < 0.05$) for total activity. In Maipo basin the behavioral responses expressed at similar levels irrespective of conductivity, with the exception of swimming behavior, which was marginally lower ($P = 0.04$) at low conductivity than at medium conductivity. The behavioral traits evaluated showed significant repeatability (Intraclass Correlation Coefficient—ICC), thought the magnitude was different among them. Swimming and pushing-up behaviors showed low repeatability as compared to walking and sheltering (S1 Table).

## Discussion

Survival and behavioral responses of *Smicridea annulicornis* showed to be sensitive to water salinity in our laboratory treatments. However, the responses were stronger in organisms whose geographic origin was the river with lower historical levels of water salinity. This result support our general prediction that geographical origin shape the responses to water salinity. Although, this result contrasts with those of Kefford et al. [35], where organisms collected from different rivers having different conductivity ranges did not exhibit differences in their salinity tolerance. Differences in tolerance of invertebrates to salinity increments has been found associated to taxonomic affiliation [36] or habitat type (e.g. lentic vs lotic, [37]). However, there is little evidence on differences in salinity responses attributable to geographic origin [38]. This is surprising considering the current relevance of salinity pollution for the conservation of freshwater biodiversity [26]. Freshwater ecosystems suffering from anthropogenic salinization have shown effects not only on community structure but also on the phenotypic expression of reproductive, morphological and physiological traits [39].

Regarding our competing hypotheses about the way in which experimental organisms respond according to their geographical origin, our results supported our second hypothesis. This hypothesis proposes that low-salinity adapted organisms should be more sensitive to experimental changes in salinity, as compared to organisms adapted to higher salinity levels. This supports that high salinity levels, within the observed ranges in Mediterranean rivers, act as a stressing factor to benthic macroinvertebrate organisms [40]. These effects could be modulated by differential osmoregulatory capabilities due to local adaptation [25]. Many species facing osmotic stress have developed adaptations to deal with salinity variation in their environment [41]. Furthermore, there is evidence of rapid evolution of osmoregulatory adaptation in aquatic organisms, which indicates that local adaptation could occur within few generations in the wild [42]. Recent findings showed these adaptation being present in benthic macroinvertebrates facing sublethal salinity levels [43, 44]. The mechanisms involve adjustment (i.e. up and down regulation) of the abundance of specialized cells (ionocytes) as well as of the abundance of transporter proteins in their membranes. Thus, the conjecture of local adaptation to explain the observed effect of geographical origin in our study organisms appears plausible, regarding the current understanding of physiological mechanisms of osmotic regulation in a variety of aquatic organisms [45–47]. Also, our experimental results are in line with field information that indicates salinity increment in rivers having a detrimental effect on biodiversity [26]. Furthermore, increased salinity has also shown to induce changes in community structure [48], and affect the phenotypic variation of the freshwater species. In this vein, it is expected that selection for tolerant variants triggers biodiversity loss, both at intraspecific and interspecific levels [25, 26].

This study aids to expand our knowledge about how the organisms respond to environmental variation, specifically to salinity, and which is the role of the historical context in shaping phenotypic responses to ecological conditions. In particular, we showed that organisms from different geographic locations responded differently to salinity, both in survival and behavior. The significant interaction effect between behavioral category and basin (Table 2) could be attributed to basin-specific environmental parameters not considered in our experimental setup, or to different microevolutionary trajectories of the populations. This highlights the potential role of local adaptation processes in shaping phenotypic and genetic variation in freshwater populations. In Chilean rivers, *S. annulinicornis* is recognized as a conspicuous species, exhibiting a wide geographical distribution with a clear genetic structure among habitats, studied at the level of allele and haplotype frequencies [31]. This suggests that genetic differentiation may explain the variation of behavioral responses observed in our study organisms across basins. This does not discard the possibility of having a set of closely related species or subspecies still unrecognized. The observed geographically—driven diversity of responses to the instantaneous level of water salinity, reflects that the maintenance of intraspecific diversity –which influence extinction risk in changing environments– rests on the conservation of geographic variation. This appeals for focusing on a landscape—level scale for planning biological freshwater conservation actions. However, our behavioral results should be interpreted with caution. As usual in laboratory experimentation, it is difficult to assess the consistency between the observed behavioral responses under controlled conditions and the animals' behavior in the field. Although we analyzed the responses of juvenile developmental stages of our model species *S. annulicornis*, it would be relevant to extend this kind of studies to assess effects also on adult stages, where reproductive outcomes could be measured. Furthermore, and since emerging adult stages are typical prey of several riparian carnivores, studying how water conditions affect adult stages of these invertebrates also allows understanding how the effects of environmental perturbations propagate between adjacent habitats that traditionally have been considered functionally apart from each other, such as river and terrestrial systems. This represents a topic of increasing interest [49–52].

## Supporting information

**S1 Table. Repeatability and responses of each behavioral trait.**
(DOCX)

## Author Contributions

**Conceptualization:** Mauricio J. Carter, Rodrigo Ramos-Jiliberto.

**Formal analysis:** Mauricio J. Carter, Matías Flores, Rodrigo Ramos-Jiliberto.

**Funding acquisition:** Mauricio J. Carter, Rodrigo Ramos-Jiliberto.

**Investigation:** Mauricio J. Carter, Matías Flores, Rodrigo Ramos-Jiliberto.

**Methodology:** Mauricio J. Carter, Rodrigo Ramos-Jiliberto.

**Project administration:** Mauricio J. Carter.

**Supervision:** Mauricio J. Carter, Rodrigo Ramos-Jiliberto.

**Validation:** Mauricio J. Carter.

**Writing – original draft:** Mauricio J. Carter, Rodrigo Ramos-Jiliberto.

**Writing – review & editing:** Mauricio J. Carter, Matías Flores, Rodrigo Ramos-Jiliberto.

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
