## [Decision Letter · Decision Letter 0]

8 Nov 2019

PONE-D-19-19484

Geographical origin determines responses to salinity of Mediterranean caddisflies

PLOS ONE

Dear Dr. Ramos-Jiliberto,

Thank you for submitting your manuscript to PLOS ONE. After careful consideration, we feel that it has merit but does not fully meet PLOS ONE’s publication criteria as it currently stands. Therefore, we invite you to submit a minor revision of the manuscript that addresses the points raised during the review process.

In your re-submission, please carefully address each of the reviewer comments. The significant difference in responses between basins was striking, raising the question about the nature of population differentiation. Please address this point in revisions, as well as more minor comments regarding clarification of details.

We would appreciate receiving your revised manuscript by Dec 23 2019 11:59PM. To enhance the reproducibility of your results, we recommend that if applicable you deposit your laboratory protocols in protocols.io, where a protocol can be assigned its own identifier (DOI) such that it can be cited independently in the future. For instructions see: http://journals.plos.org/plosone/s/submission-guidelines#loc-laboratory-protocols

We look forward to receiving your revised manuscript.

Kind regards,

Rachel A Hovel

Academic Editor

PLOS ONE

Journal Requirements:

1. In your Methods section, please provide additional location information of the study areas, including geographic coordinates for the data set if available.

Reviewers' comments:

Reviewer's Responses to Questions

**Comments to the Author**

1. Is the manuscript technically sound, and do the data support the conclusions?

Reviewer #1: Yes

Reviewer #2: Yes

2. Has the statistical analysis been performed appropriately and rigorously? 

Reviewer #1: Yes

Reviewer #2: Yes

3. Have the authors made all data underlying the findings in their manuscript fully available?

Reviewer #1: No

Reviewer #2: Yes

4. Is the manuscript presented in an intelligible fashion and written in standard English?

Reviewer #1: Yes

Reviewer #2: Yes

5. Review Comments to the Author

Reviewer #1: According to the instructions to authors, the authors either need to place the data in a public repository, where URL, accession numbers or DOIs can be cited, or can be contained in Supporting Information files. The current statement says that the data will be available upon request.

The paper needs a little editing, but, per instructions, I did not note specific edits, except that in the second sentence of the Introduction, the verbs should be "disrupts" and "threatens".

The authors might consider discussing how a species might adapt to differences in salinity across its range. At higher taxonomic levels, adaptation to different salinities can involve small mutations to the genes for the ion transporter genes that affect influx and efflux rates. In anadromous or catadromous species, such as salmonids and eels, adaptation may involve up-regulation or down-regulation of these ion transporters or of entire ionocytes.

Reviewer #2: Carter, Flores, Ramos-Jiliberto

Geographical origin determines responses to salinity of Mediterranean caddisflies

This study looks the importance of source (river basin of origin) when assessing behavioral and survivorship responses of larval caddisflies (Smicridea annulicornis) when exposed to elevated salinity. This species appears to be widely distributed across Chile and Argentina, and expresses significant genetic differentiation within a basin, and across basins. Larvae were collected from three basins: Choapa (low conductivity), Maipo (medium conductivity) and Maule (high conductivity), and based on ambient conductivities, they tested three experimental conductivity levels: low (180 μS/cm), medium (500 μS/cm) and high (1400 μS/cm) different salinity levels were prepared with aquarium sea salt dissolved in distilled water. Response variables measured were time to death, and behavior (swimming, sheltering, walking, pushing up). This assessment of behavioral responses (i.e., non-lethal) was an interesting addition to the standard survival study. The experiments found significant differences between basins – big differences. This is most unusual and suggests measurable local adaptation or acclimation.

One factor that is only lightly addressed is this species in this region exhibits significant genetic differentiation among basins (Sabando et al. 2011) – suggesting either local adaptation with little or no gene exchange, or possibly some closely related species or subspecies that are unrecognized and responding to a strong habitat template.

Specific comments;

I assume the experiments were conducted at 20 C, same as the acclimation temperature.

Survival time – Table 1 indicates that basin matters, and that the effect of conductivity was not significant (p=0.08), but the results present a bunch of antidotal assessements of conductivity effects. Table 1 results and the Survival results need to be in agreement – at least justify the details by calling the factorial result “nearly significant” and not “marginal”.

In the behavioral tests, why do you think the BC x Basin was significant? Why would behavioral responses differ between basins within a species?

In the behavioral tests, do you believe behavioral responses are in some what adaptive with regard to salinity?

The results presented here make one question standard bioassay results – where was the original test population from? Have laboratory conditions changed the test organisms?

6. PLOS authors have the option to publish the peer review history of their article (what does this mean?). If published, this will include your full peer review and any attached files.

Reviewer #1: No

Reviewer #2: No

---

## [Author Response · Author response to Decision Letter 0]

28 Nov 2019

PONE-D-19-19484

Geographical origin determines responses to salinity of Mediterranean caddisflies

PLOS ONE

RESPONSE TO REVIEWERS (POINT-BY-POINT)

Journal Requirements:

Please ensure that your manuscript meets PLOS ONE's style requirements, including those for file naming. Response: Checked

1. In your Methods section, please provide additional location information of the study areas, including geographic coordinates for the data set if available.

Response: Done as requested. L 79-80

Response: No permits are required, according to national regulations. We included this information in the text. L 80-82

Response: We uploaded the data to the dryad repository. We included this information in the cover letter as requested.

Review Comments to the Author

Reviewer #1: 

According to the instructions to authors, the authors either need to place the data in a public repository, where URL, accession numbers or DOIs can be cited, or can be contained in Supporting Information files. The current statement says that the data will be available upon request.

Response: We uploaded the data to the dryad repository. We included this information in the cover letter as requested.

The paper needs a little editing, but, per instructions, I did not note specific edits, except that in the second sentence of the Introduction, the verbs should be "disrupts" and "threatens".

Response: Corrected. L 4-5

The authors might consider discussing how a species might adapt to differences in salinity across its range. At higher taxonomic levels, adaptation to different salinities can involve small mutations to the genes for the ion transporter genes that affect influx and efflux rates. In anadromous or catadromous species, such as salmonids and eels, adaptation may involve up-regulation or down-regulation of these ion transporters or of entire ionocytes.

Response: Corrected. In the 2nd paragraph of the Discussion section we now talk about rapid evolution and mechanisms of ion regulation in aquatic organisms, including macroinvertebrates. L 210-227

Reviewer #2: 

One factor that is only lightly addressed is this species in this region exhibits significant genetic differentiation among basins (Sabando et al. 2011) – suggesting either local adaptation with little or no gene exchange, or possibly some closely related species or subspecies that are unrecognized and responding to a strong habitat template.

Response: We extend our discussion about the empirical evidence of genetic structure in our study species on the study sites, and also include the reviewer’s idea of unrecognized species or subspecies. L 237-242

Specific comments;

I assume the experiments were conducted at 20 C, same as the acclimation temperature.

Response: Yes. This information was included in the revised version. L 102-104

Survival time – Table 1 indicates that basin matters, and that the effect of conductivity was not significant (p=0.08), but the results present a bunch of antidotal assessments of conductivity effects. Table 1 results and the Survival results need to be in agreement – at least justify the details by calling the factorial result “nearly significant” and not “marginal”.

Response: We corrected the text following the reviewer's suggestions. L 148

In the behavioral tests, why do you think the BC x Basin was significant? Why would behavioral responses differ between basins within a species?

Response: Done. We added a phrase in which we –with caution-- mention possible causes. L 232-235

In the behavioral tests, do you believe behavioral responses are in some what adaptive with regard to salinity?

Response: Our results do not allow inferring that behavioral responses are adaptive respect to salinity. We guess that the previous response covers this aspect too.

The results presented here make one question standard bioassay results – where was the original test population from? Have laboratory conditions changed the test organisms?

Response: Corrected. We commented on the limitation of lab experimentation. 250-253

---

## [Editor Report · Decision Letter 1]

13 Dec 2019

Geographical origin determines responses to salinity of Mediterranean caddisflies

PONE-D-19-19484R1

Dear Dr. Ramos-Jiliberto,

We are pleased to inform you that your manuscript has been judged scientifically suitable for publication and will be formally accepted for publication once it complies with all outstanding technical requirements.

With kind regards,

Rachel A Hovel

Academic Editor

PLOS ONE

Additional Editor Comments (optional):

This revision has sufficiently addressed reviewer comments and outstanding concerns on this manuscript, and I am pleased to accept it. Please make sure to thoroughly proofread the manuscript before submission. For example, Line 71 now reads: "We conducted this study usin as a model speciesgthe...". It will be your responsibility to identify and remedy typographical or other errors.
---

## [Editor Report · Acceptance letter]

19 Dec 2019

PONE-D-19-19484R1 

Geographical origin determines responses to salinity of Mediterranean caddisflies 

Dear Dr. Ramos-Jiliberto:

I am pleased to inform you that your manuscript has been deemed suitable for publication in PLOS ONE. Congratulations! Your manuscript is now with our production department. 

With kind regards,

on behalf of

Dr. Rachel A Hovel 

Academic Editor

PLOS ONE